# Elucidating T Cell and B Cell Responses to SARS-CoV-2 in Humans: Gaining Insights into Protective Immunity and Immunopathology

**DOI:** 10.3390/cells11010067

**Published:** 2021-12-27

**Authors:** Aaruni Khanolkar

**Affiliations:** 1Department of Pathology, Ann and Robert H. Lurie Children’s Hospital of Chicago, 225 East Chicago Avenue, Box 82, Chicago, IL 60611, USA; AKhanolkar@luriechildrens.org; Tel.: +1-312-227-8073; 2Department of Pathology, Northwestern University, Chicago, IL 60611, USA

**Keywords:** SARS-CoV-2, T cells, B cells, COVID-19, immunological memory, immunopathology

## Abstract

The SARS-CoV-2 pandemic is an unprecedented epochal event on at least two fronts. Firstly, in terms of the rapid spread and the magnitude of the outbreak, and secondly, on account of the equally swift response of the scientific community that has galvanized itself into action and has successfully developed, tested and deployed highly effective and novel vaccines in record time to combat the virus. The sophistication and diversification of the scientific toolbox we now have at our disposal has enabled us to interrogate both the breadth and the depth of the immune response to a degree that is unparalleled in recent memory. In terms of our understanding of what is critical to contain the virus and mitigate the effects the pandemic, neutralizing antibodies to SARS-CoV-2 garner most of the attention, however, it is essential to recognize that it is the quality and the fitness of the virus-specific T cell and B cell response that lays the foundation and the backdrop for an effective neutralizing antibody response. In this report, we will review some of the key findings that have helped define and delineate some of the essential attributes of T and B cell responses in the setting of SARS-CoV-2 infection.

## 1. Introduction

The beginning of each decade of the 21st century has been punctuated by a novel coronavirus (CoV) outbreak [1,2,3,4]. Although the SARS-CoV (2002–2003) and MERS-CoV (2012) outbreaks were associated with much higher mortality rates when compared to SARS-CoV-2, they were relatively restricted in terms of scale and magnitude [1,2,3]. The ability of SARS-CoV-2 to establish sub-clinical, asymptomatic infections in large swaths of the population promoted the “silent spread” of this novel zoonotic CoV and ultimately propelled it to become the most significant public health emergency since the Spanish Flu pandemic of 1918 [5]. However, what is clearly different now when compared to the 1918 pandemic are the advances made in science, especially in the areas of epidemiology, virology, immunology and computational biology, all key scientific disciplines that are particularly germane to studying outbreaks of this nature. These advances have enabled us to accomplish in-depth and highly accurate assessments to rapidly identify novel pathogens, perform high-throughput analyses to gauge pathogen exposure at molecular, serologic and cellular levels, as well as design fairly accurate prediction models relating to the evolution of the pathogen and the host’s ability to mediate protection against emerging variants of concern.

## 2. T Cell Responses in the Setting of SARS-CoV-2 Infection

Our immune system has evolved as a multi-component, hierarchical biologic entity with clearly defined divisions of labor [6]. Within this intricate network, one of the functions the T cells discharge is the targeted eradication of infected cells, and they accomplish this task both in the setting of a primary infection or vaccination as well as re-infection by the same pathogen, with virus-specific memory T cells playing a prominent role in this latter instance [7]. T cells also provide critical “help” to B cells to generate a potent and specific humoral immune (Ab) response that primarily operates to eliminate extracellular pathogens [7].

A plethora of published studies have examined the role of T cells, neutralizing antibodies and B cells in the context of SARS-CoV-2 infection. Some of the findings from these studies seem contradictory, perhaps reflecting differences in the patient populations (including ethnicity/genetic background) and cohort sizes examined, tissues sampled for immunological analyses (peripheral blood versus airway T cells), and also the populations of T cells interrogated in the studies (antigen-specific or bulk T cell responses). Nonetheless, collectively these studies highlight the value of performing in-depth analyses of T cell responses in individuals exposed to SARS-CoV-2 and/or recipients of the COVID-19 vaccine (Figure 1).

### 2.1. Correlating T Cell Responses with COVID-19 Disease Phenotypes

Several studies have defined a transient diminution (lasting 2–3 weeks following onset of symptoms) in the proportion of circulating T cells in SARS-CoV-2 infected patients [8,9,10,11]. Specifically, a marked depletion of CD4 effector memory T cells (Tem) of the Th1 and Th17 lineages, as well as CD8 Tem and Vγ9Vδ2 T cell subsets and relative preservation of Th2 cells, regulatory T cells (Tregs) and Vδ1^+^ γδ T cells has been reported [12]. A direct consequence of this effect is an enhanced neutrophil to lymphocyte ratio in the blood, and the degree to which this ratio is amplified appears to be predictive of disease severity, with at least one study reporting that an elevated ratio of immature neutrophils to CD8 T cells as well as γδ T cells displays high sensitivity and specificity for predicting hypoxia onset as well as pneumonia [11,13]. This phase of transient lymphopenia might reflect activation-induced cell death but could also be suggestive of the initial egress of T cells from the blood to the tissue sites experiencing active viral replication, and the subsequent re-establishment of homeostasis after the pathogen is successfully contained [11,12]. This latter notion is supported by the fact that the recovery of circulating T cell numbers is characterized by the emergence of CD127^+^ Tem and central memory (Tcm) CD4 T cells, as well as terminally differentiated CD57^+^ CD8 T cells [11]. In this context, the retention of Th2 cells and the initial depletion of Tem from the circulation makes biological sense, given the protective potential of Tem in the setting of viral infections. The perceived retention of Tregs in the periphery could ostensibly also allow the host to effectively tackle the virus in the tissue, although an unintended consequence of this might be an unbridled T cell response in the tissues, such as the lungs, which could potentially contribute to immunopathology in individuals predisposed to dysregulated immune responses.

The progression to severe disease that occurs in some SARS-CoV-2 infected individuals appears to be triggered by a dysregulated response involving multiple but linked elements of the host’s immune system [8,12,47,48,49]. Given that the immune system has not encountered this specific pathogen previously, it lacks a reliable historical record or playbook so to speak, that it can depend on to optimally control this virus. Hence, the immune dysregulation and attendant immunopathology could entail overcompensation from one or several arms of the immune system, which could include the adaptive immune compartment, in response to molecular cues perceived as exerting sub-optimal control over the invading pathogen by effectors that operate at the frontline of the primary immune response. Additionally, data from murine models of a natural, respiratory CoV (Mouse Hepatitis Virus-1) infection suggest that immunopathology could be an inherited trait underpinned by background genes other than the major histocompatibility complex (MHC) genes that regulate T cell responses [50,51].

Sophisticated analytic tools, such as Uniform Manifold Approximation and Projection (UMAP), have been utilized in several studies to define immunological signatures (immunotypes) that correlate with clinical outcomes in SARS-CoV-2 infected individuals. Mathew et al. linked highly activated, circulating CD4 T cells with severe disease in patients hospitalized with COVID-19, while a restrained pattern of T and B cell activation was observed in hospitalized patients with a less severe disease course, suggesting T cell mediated immunopathology as a component of severe disease [9]. Other investigators that have also evaluated T cell responses in SARS-CoV-2 infected individuals have reported a few contrasting findings based on observations of divergent clinical trajectories and disparities in T cell responses that segregate with age and gender [14,15]. In general, most children experience a relatively milder disease course, while individuals over the age of 65 years display a propensity to develop severe complications related to COVID-19, which may be attributable, at least in some measure, to an uncoordinated adaptive immune response that is underpinned by a relative paucity of naïve T cells [15,16,17,18]. These observations imply a link between the degree of T cell “experience”, as well as aging-related impairments in the immune-regulatory mechanisms involved in SARS-CoV-2 infection-associated disease outcomes. Gender-based investigations have shown that adult males with COVID-19 fare worse compared to adult females, and a hallmark of this difference in disease course is a relative insufficiency of activated (CD38^+^, HLA-DR^+^), as well as terminally differentiated (PD-1^+^ and TIM-3^+^) and IFNγ-producing CD8 T cells in adult males with COVID-19 [14]. Auto-antibodies to Type I IFN that can enhance COVID-19 related morbidity are also observed at a much higher frequency among men than women [19]. On the other hand, adult female subjects that are projected to fare poorly after exposure to SARS-CoV-2 experience an over-exuberant innate immune serum cytokine response, with TNFSF10 and IL-15 being notably over-expressed [14]. Taken together, these findings are consistent with previous reports of gender-related differences in the potency of lymphocyte responses, and a corollary to this point is the higher prevalence of autoimmune diseases in females [52,53,54,55]. Although several groups have reported autoimmune responses in COVID-19 patients, it appears that acute respiratory failure irrespective of the initial insult can induce a diverse set of autoimmune responses [56,57]. Furthermore, given that males are more likely to suffer adverse outcomes following SARS-CoV-2 infections, this might account for the higher frequency of auto-Ab to Type I IFN observed in male patients.

In addition to examining bulk T cell populations, a number of reports have also carefully dissected out the SARS-CoV-2 specific T cell response, characterizing both the phenotypic properties and functional attributes of the responder cells early after onset of symptoms, as well as at later time points (≥6 months post-infection), and these assessments have revealed that these virus-specific T cell responses contract with a half-life of about 3–5 months [15,20,21,22,23,24,25,58]. By combining sequence homology data for SARS-CoV-2 with other human CoV, and state of the art bioinformatic approaches, Grifoni et al. identified SARS-CoV-2 peptide epitopes that were predicted to bind the twelve most prominent HLA class I-A and -B alleles (covering ~85% of the general population) as well as multiple HLA class II alleles across different ethnicities (haplotypes) and allelic variants [20]. These peptide pools were then incorporated as stimulants in activation-induced marker (AIM) and intracellular cytokine secretion assays (ICS) to induce virus-specific CD4 and CD8 T cell responses [20]. Based on these analyses, the target distribution of the virus-specific CD4 T cell response was predominantly focused on the structural proteins (spike protein (27%), membrane protein (21%) and nucleocapsid protein (11%)), while the responses directed against the non-structural proteins (nsp) three, four and ORF8 comprised a total of 15% [20]. Similar frequencies were noted for the virus-specific CD8 T cells targeting the spike, membrane and nucleocapsid proteins, while the non-structural proteins that were targeted included nsp6, ORF8 and ORF3a, and this fraction comprised 32% of the CD8 T cell response [20]. An analogous approach to evaluating SARS-CoV-2 specific T cell responses has revealed that the HLA-B*07:02 restricted nucleoprotein epitope N105-113 is one of the most dominant CD8 T cell epitopes, and its dominance is underwritten by a high naïve precursor frequency (evident in both SARS-CoV-2 infected individuals as well as pre-pandemic PBMC specimens) and a diverse set of TCRαβ clonotypes [59]. In contrast, the SARS-CoV-2 epitopes restricted to the HLA-A*02:01 allele (which is highly prevalent in the Caucasian population), S269-277 and Orf1ab3183-3191, evoked relatively tepid CD8 T cell responses, that were mostly lacking the expression of activation markers such as CD38, HLA-DR, PD-1 and CD71, and displayed phenotypic profiles that segregated with naïve, stem cell memory and central memory T cells rather than effector memory T cells [60]. As the HLA-B*07:02 and HLA-A*02:01 allele frequencies in the human population are quite high, large scale prospective studies that track SARS-CoV-2 specific CD8 T cell responses restricted to these alleles in conjunction with the clinical course could yield a wealth of valuable immunologic data relevant to the pandemic [61].

In an attempt to define the T cell responses to SARS-CoV-2 on a scale that is partially commensurate with scale of the pandemic, one study utilized an in-silico approach to examine the SARS-CoV-2 peptidome and predict the binding affinity of the SARS-CoV-2 peptides to almost 10,000 different HLA class I and II alleles, thereby identifying putative epitopes in over 90% of the population, covering almost 60 countries [62]. A separate report adopted an agnostic in vitro strategy that applied mass-spectrometry to examine peptide–HLA complexes from SARS-CoV-2 infected cells at different time points following infection, which facilitated the identification of novel HLA class I restricted peptide epitopes derived from out-of-frame open reading frames (ORFs) that induced more robust CD8 T cell responses than those observed with canonical peptides [63]. This attempt highlights the potential advantage of unbiased epitope screening approaches over traditional methods that are reliant on evaluating T cell responses restricted to well-known HLA-supertypes, which might under-estimate the quality and quantity of the CD8 T cell responses to SARS-CoV-2 [61].

Mirroring some of the observations noted with bulk T cell analyses, studies attempting to define correlative relationships between antigen-specific T cells and the clinical disease course have also reported some contrarian viewpoints. For instance, Kusnadi et al. and Sekine et al. have reported that a predominance of Granzyme B (Grz B)^+^ SARS-CoV-2 specific CD8 T cells correlated with a higher degree of morbidity, and, additionally, Kusnadi et al. observed that a surplus of exhausted virus-specific CD8 T cells were identified in patients with milder but not serious disease [24,26]. However, Moderbacher et al. presented slightly opposing lines of evidence, demonstrating a robust positive association between mild disease and total numbers of IFNγ-producing SARS-CoV-2 specific CD8 T cells, as well as total numbers of SARS-CoV-2 specific CD4 T cells and circulating T follicular helper (Tfh) cells in the acute phase of the illness [15]. This study also alluded to the possibility that elevated serum CXCL10 levels might serve as a potentially informative surrogate acute phase marker that identifies sub-optimal SARS-CoV-2 specific CD4 and CD8 T cell responses [15]. Along similar lines, other investigators have also reported higher frequencies of SARS-CoV-2 specific polyfunctional CD8 T cells in patients with mild disease [27]. Perhaps differences in the assay platforms that are utilized in some of these studies could account for the disparity in their findings. Kusnadi et al. relied on transcriptomic analyses while the read-out in the Moderbacher study involved conventional cell-mediated protein expression. Related to this issue, disparity between IL-6 mRNA and protein levels has been described in the setting of SARS-CoV-2 infection [47].

Assessments of SARS-CoV-2 specific memory T cell subsets in tissues of seropositive decedents demonstrate that cells can be identified in the bone-marrow (BM), spleen and gut-associated lymph nodes (LN), in addition to the lungs and lung-draining LN up to at least six months post-infection [64]. In this report, CD4 T cells existed primarily as effector memory T cells (Tem) in these tissues, while the Tem and CD45RA^+^Tem (TemRA) subsets predominated in the CD8 T cell pool, and the canonical tissue-resident memory (Trm) CD4 and CD8 T cells that co-express CD69 and CD103 were primarily observed in the lungs [64]. Published evidence also suggests that younger patients and those that experience a more favorable course of the disease harbor higher frequencies of activated, tissue (airway) resident T cells that co-express CD69, CD103, PD-1 and HLA-DR [28]. This association between reactive airway resident T cells and disease course is reported to be a more compelling measure of disease severity than standard assessments, such as the Sequential Organ Failure Assessment (SOFA) score, and the and PaO_2_:FiO_2_ (P:F) ratios that are utilized to stratify the clinical status of patients experiencing Acute Respiratory Disease Syndrome (ARDS) [28]. Interestingly, the lungs of patients that experience severe COVID-19 can harbor clonally expanded Th17 cells that produce copious amounts of IL-17A and Granulocyte Monocyte-Colony Stimulating Factor (GM-CSF), which offers a potential opportunity for therapeutic interventions that can impede GM-CSF signaling by administering anti-GM-CSF receptor monoclonal antibodies, such as mavrilimumab or lenzilumab [29,30,31].

### 2.2. Cross-Reactive T Cells

Another finding that has surfaced is the identification of T cells in SARS-CoV-2 unexposed, healthy donors that respond to SARS-CoV-2 peptides in the AIM and ICS assays [20,21,22,23,24,25,32]. This is not altogether unexpected, given that certain strains of CoV (HKU1, OC43, 229E and NL63) are endemic in the human population and cause ~20% of upper respiratory tract infections in adults, and in certain regions of the world, cultural and occupational practices might increase the odds of potential exposure to animal betacoronaviruses [3]. This endemicity has quite likely induced and sustained memory T (and B) cell responses, targeting the products of several structural and non-structural CoV genes that display varying degrees of sequence identity among the different human CoV strains. As far as the SARS-CoV-2 spike protein is concerned, cross-reactive CD4 T cell responses are generally focused toward the epitopes localized at the C-terminal end of the protein, due to greater sequence homology with the spike protein of the endemic human CoV (a ≥ 67% amino-acid sequence homology among defined HLA-class II epitopes is a reliable cut-off for predicting CD4 T cell cross-reactivity) [21]. The same study that initially defined the T cell responses to SARS-CoV-2 demonstrated that almost 50% of the unexposed, healthy subjects included in their investigation displayed CD4 T cell responses that reacted against a wide range of SARS-CoV-2 peptides utilized in their assays [20]. Roughly 23% of the responding CD4 T cells reacted to the SARS-CoV-2 spike protein, and about 54% of the CD4 T cells responded to stimulation by nsp14, nsp4 and nsp6 [20]. Cross-reactive CD8 T cell responses were less prominent and were only detected in 20% of the healthy, unexposed study subjects [20]. Differences between CD4 and CD8 T cells in terms of the stringency requirements with respect to epitope sequence homology might account for this difference [32]. Other investigators have also reported ranges between 30–50% of unexposed, healthy donors displaying SARS-CoV-2 specific cross-reactive T cell responses to the SARS-CoV-2 spike protein, membrane and nucleoprotein, as well as nsp7 and nsp13 [20,21,22,23,24,25,32]. Once again, differences in the study populations evaluated can generate findings that are seemingly at variance from those reported by others. For instance, a Swedish study reported that nucleoprotein-specific cross-reactive T cells were conspicuously absent in their cohort of SARS-CoV-2 unexposed individuals, while other publications have described the viral nucleoprotein-specific T cells as being a prominent subset of the cross-reactive pool of T cells [24,25]. An important issue that remains unresolved thus far is whether these pre-existing cross-reactive T cells will play a protective or pathologic role once the hitherto uninfected host is subsequently exposed to SARS-CoV-2 [32,33,34]. In one retrospective study, individuals with recent exposure to endemic human CoV endemic experienced a milder disease course following subsequent SARS-CoV-2 infection, and a subsequent study demonstrated that human CoV specific, SARS-CoV-2 cross-reactive CD4 T cells can be rapidly recruited into the immune response following natural SARS-CoV-2 infection and immunization, while other studies have demonstrated the sub-optimal cytotoxic activity and polyfunctional potential of endemic human CoV derived SARS-CoV-2 cross-reactive T cells [35,36,37,65]. Carefully designed prospective studies evaluating the clinical outcomes of SARS-CoV-2 infection in SARS-CoV-2 unexposed individuals harboring cross-reactive T cells will be necessary to provide conclusive evidence relating to the protective and/or pathological potential of cross-reactive T cells.

### 2.3. T Cell Responses in Multi-Inflammatory Syndrome in Children (MIS-C)

While a majority of the publications have described T cell responses in adults, a few studies have attempted to define the immune signature in children with COVID-19, including pediatric subjects that develop Multi-Inflammatory Syndrome in Children (MIS-C) [10,66,67,68]. An early description of the peripheral bulk immune signature in MIS-C revealed a picture not unlike that of SARS-CoV-2+ patients without MIS-C, with transient T and B cell lymphopenia, activated (CD64hi) neutrophils and monocytes, upregulated HLA-DR surface expression on CD4 T cells and γδ T cells, and conversely reduced HLA-DR and CD86 expression on antigen-presenting cells [10]. A subsequent study also noted that there were no major differences in the immunotypes of SARS-CoV-2 infected pediatric patients with and without MIS-C [66]. Both sets of pediatric patients had reduced proportions of naïve CD4 T cells and T follicular helper (Tfh) cells, and an over-representation of effector and central memory CD4 T cells, as well as terminally differentiated, senescent CD4 T cells that expressed CD57 [66]. The only notable variance between the MIS-C and non-MIS-C patients in this study was the greater deficit in the CD8 T cell population in the MIS-C group [66]. Others have noted additional differences; for instance, MIS-C patients harbor significantly higher frequencies of circulating PD-1^+^CD4 T cells and PD-1 and CD39 co-expressing CD8 T cells when compared to non-MIS-C patients, alluding to the possibility that antigen persistence may be a feature of MIS-C [67]. Additionally, consistent with the vascular involvement observed in MIS-C patients, highly activated, proliferating CD8 T cells expressing the fractalkine receptor, CX3CR1, that facilitates the vascular surveillance capability of CD8 T cells, are prominently represented in MIS-C patients [67]. Significant skewing of the T cell receptor (TCR) repertoire in favor of *TRBV11-2* in both CD4^+^ and CD8^+^ memory T cells, as well as the enhanced expression of Granzyme A in CD45RA^+^ effector memory CD8 T cells (TemRA) and *ITGB7* in memory CD8 T cells have also been noted in MIS-C patients [68]. These latter observations also provide some potential clues into the pathogenesis of this unique hyper-inflammatory disorder, since *TRBV11-2* has been associated with autoreactive T cells that recognize non-peptide antigens in the context of non-classical HLA molecules, and Granzyme A can contribute to both cytotoxicity and inflammation by virtue of its ability to cleave pro-IL-1, and ITGB7 is a gut-homing integrin molecule subunit (abdominal involvement is a key feature of the clinical presentation in MIS-C patients) [68]. At least one study has defined virus-specific CD4 T cell responses in pediatric subjects with and without MIS-C, reporting that the viral spike-specific IFNγ^+^ CD4 T cell responses did not vary between the two groups, and that overall, adults displayed stronger viral spike-specific IFNγ^+^ CD4 T cell responses than children [69]. Blood volume requirements for performing the comprehensive profiling of virus-specific T cell responses could likely contribute to the paucity of these reports in pediatric subjects.

### 2.4. T Cell Responses in COVID-19 Vaccine Recipients

Given that the pandemic currently continues unabated and the SARS-CoV-2 vaccines were deployed in the general population only a few months back, descriptions of vaccine-induced SARS-CoV-2 specific T cells have predominantly focused on acute and early memory phase responses. Recipients of the Pfizer (BNT162b2) and Moderna (mRNA-1273) vaccines generate spike-specific CD4 T cells that recognize multiple peptide epitopes from the wild-type SARS-CoV-2, and also recognize, to an equal degree, some epitopes that are altered in the B.1.1.7 and B.1.351 variants [38]. These mRNA vaccines also appear to strongly boost CD4 T cell responses to the spike protein of the endemic HCoV-NL63 virus [38]. Utilizing peptide-MHC multimers, intracellular cytokine staining and ELISPOT to track BNT162b2 vaccine-induced spike-specific CD4 and CD8 T cells, Sahin et al. reported the presence of effector memory T cells that elaborated predominantly IFNγ (CD8 T cells) or IFNγ+IL-2+ (CD4 T cells) [39]. Spike-specific CD8 T cells accounted for 0.02–2.92% of the total circulating CD8 T cells a week following the booster dose, and these cells could still be tracked eight weeks later (0.01–0.28% of the total circulating CD8 T cells) [39]. Analyses from subjects that were exposed to the 2002-2003 SARS-CoV have demonstrated that SARS-CoV-specific memory T cell responses are maintained long term and are detectable up to 17 years after initial exposure [25]. This degree of durability of the T cell response is encouraging and engenders a sense of cautious optimism about the long-term preservation of potentially protective immune responses following both natural exposure to SARS-CoV-2 and receipt of the SARS-CoV-2 vaccines. Future studies that elucidate the phenotypic and functional attributes of long-term memory T cell responses to SARS-CoV-2 will be necessary and particularly informative in this regard. The ability of SARS-CoV-2 to adapt to its host environment and evolve fairly rapidly will pose a challenge to vaccinologists, and perhaps spur the development of newer vaccines that can incorporate broader adaptive immune responses, targeting better conserved viral determinants (for example, the nucleoprotein), where immune selection-induced mutations could strongly compromise viral replication fitness [70,71].

### 2.5. Adoptive Transfer Therapy Using SARS-CoV-2 Specific T Cells

Reports have begun to emerge that demonstrate the potential utility of adoptive transfer therapy using ex-vivo expanded SARS-CoV-2 specific T cells [37,40,41,42,72,73]. One of the earliest publications in this context utilized a good manufacturing practice (GMP)-compliant methodology to, in vitro, stimulate and expand virus-specific T cells from convalescent donors using SARS-CoV-2 peptides for the purpose of adoptive immunotherapy aimed at immunocompromised patients exposed to SARS-CoV-2 [40]. A subsequent clinical trial demonstrated the safety and feasibility of adoptively transferring purified CD45RA^−^ T cells, which were obtained from SARS-CoV-2 convalescent donors, into partially HLA-matched recipients diagnosed with moderate to severe COVID-19 [41]. Given that patients with severe COVID-19 are often treated with immune suppressants, ex vivo expanded SARS-CoV-2 specific T cells can be additionally manipulated by CRISPR-Cas9 gene-editing methods to render them resistant to glucocorticoids by inactivating the glucocorticoid receptor gene (*NR3C1*) [42]. Evidence of imbalance in the regulatory T cell (Treg)/Th17 cell ratio has also been described in patients that experience respiratory failure in the setting of SARS-CoV-2 infection, hence adoptive immunotherapy approaches that utilize Tregs to mitigate SARS-CoV-2 associated hyper-inflammation are also being explored [43,44,45,46].

## 3. B Cell Responses to SARS-CoV-2

Broadly speaking, the initial step in the context of humoral immunity following exposure to a viral pathogen or a vaccine directed against a virus is the induction of a transient, extrafollicular antigen-specific plasmablast response that produces low-affinity antibodies directed against that invading pathogen [74,75,76]. Subsequently, a more intricate sequence of immunological events unfold in the secondary lymphoid tissues in the form of a germinal center (GC) reaction (involving B cells and Tfh cells) that lays the foundation for establishing long-term protective B cell immunological memory (Figure 2) [74,75,76]. Protective humoral immune memory has two major cellular components, comprising of long-lived, bone marrow resident, high-affinity antibody-secreting plasma cells, and quiescent, circulating memory B cells that serve as immune sentinels by performing critical immune-surveillance functions (Figure 2) [74,75,76]. Some of the memory B cells that preferentially express the T-box transcription factor (T-bet) and Eomesodermin, as well as surface molecules such as CD80, CD180 and Transmembrane Activator and CAML Interactor (TACI), are intrinsically programmed to rapidly differentiate into long-lived, high-affinity antibody secreting plasmablasts when the antigen is re-encountered [67,74,75,76,77,78,79].

The most commonly evaluated correlate of protective immunity induced by a natural viral infection or a vaccine directed against viral pathogens is the induction of a robust and durable neutralizing antibody (NAb) response [80,81]. The current SARS-CoV-2 pandemic-related efforts directed towards assessing the real-world effectiveness of the SARS-CoV-2 vaccines are also similarly aligned with this immunologic principle [39,76,82,83,84,85,86,87,88,89,90,91]. NAb responses operate by hindering the engagement of the virus with its receptor on the host cell [80,81]. In the context of SARS-CoV-2 infection, NAbs of multiple isotypes (IgM, IgA and IgG) block the interaction between the viral spike glycoprotein and the host cell-associated Angiotensin Converting Enzyme-2 (ACE-2) receptor [92,93]. It is generally understood that high titer NAbs are, on the whole, protective, and an ideal NAb response should mediate protection against a wide breadth of related viruses and emerging variants of concern, and a lot of work is currently being poured into deciphering and identifying such “super” antibodies in the context of the current pandemic [94]. However, it is also becoming evident that NAbs to SARS-CoV-2 can wane over time and some NAbs may paradoxically enhance SARS-CoV-2 infection by promoting syncytium formation [74,75,92,95,96,97], and antibody-mediated enhancement of SARS-CoV-2 infection-associated disease has been described [98]. In contrast to NAbs, SARS-CoV-2 specific memory B cells do not experience a similar degree of progressive diminution and can serve as a readily available source of replenishable protective Ab responses if needed [74,75]. However, SARS-CoV-2 infection-induced memory B cells can display temporal alterations in immunodominance hierarchy that shifts away from spike-specific responses in favor of the nucleoprotein and ORF8-directed responses, highlighting the importance of vaccinations to undergird and preserve protective spike-specific B cells and Ab responses [99].

### 3.1. Detecting Ag-Specific B Cells in SARS-CoV-2 Infected Individuals

In recent years reagents have been engineered to interrogate Ag-specific B cell responses [100]. The design of these reagents is conceptually related to the peptide-MHC tetramer technology, utilized to evaluate Ag-specific T cell responses, and in the case of Ag-specific B cells, biotinylated antigens are multimerized using streptavidin-fluorochrome conjugates [77,100,101]. B cells can display a greater tendency to non-specifically bind to conformationally intact protein molecules and, given that the frequency of Ag-specific memory B cells can range from 0.008 to 0.1% of total B cells, particular attention has to be paid to enhance the specificity of the measurement and minimize the detection of non-specifically binding B cells [77,100,101]. Some of the approaches that have been adopted to achieve better specificity involve utilizing non-protein polymer fluorochromes, coupling two different fluorescent tetramer constructs for the same Ag (“double-discrimination”; only the B cells that bind to both tetramers are considered true Ag-specific B cells) and including a decoy streptavidin-fluorochrome probe lacking the biotinylated antigen [77,101].

**Figure 2 cells-11-00067-f002:**
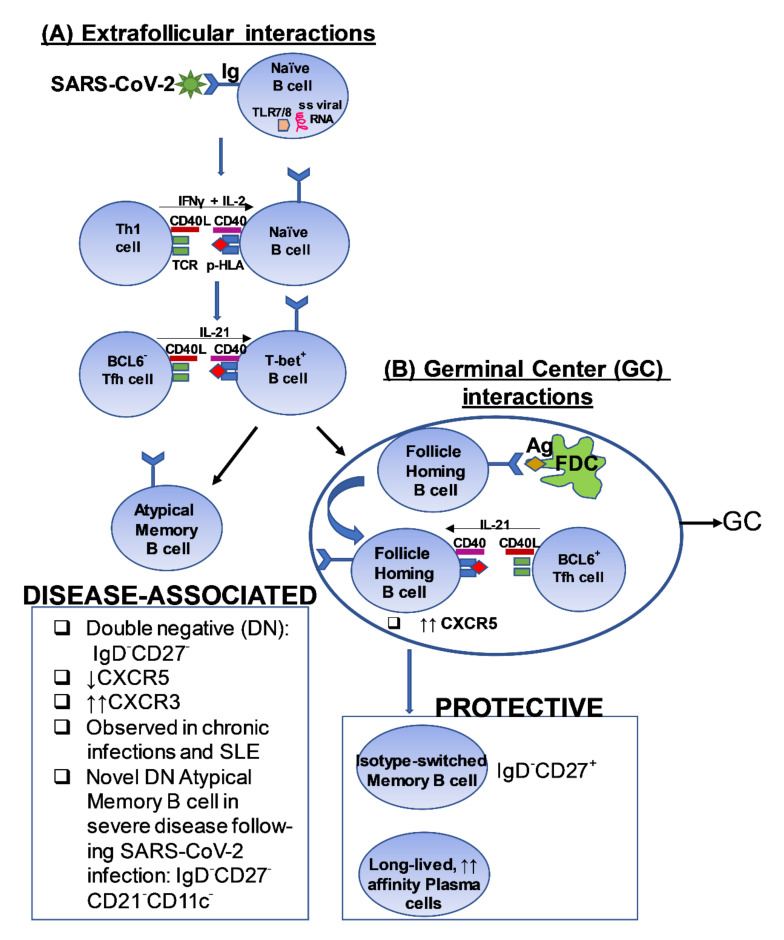
A proposed model depicting a potential pathway of cross-talk between T cells and B cells in the context of SARS-CoV-2 infection [102,103,104,105,106]. The figure outlines a putative pathway that induces the generation of (**A**) Disease-associated B cells observed in the setting of severe disease and, (**B**) Protective B cell responses, following exposure to SARS-CoV-2. FDC: Follicular dendritic cell; TLR: Toll-like receptor; TCR: T cell receptor; p-HLA: peptide-Human leukocyte Antigen complex; BCL6; B cell lymphoma 6 protein; CXCR5: C-X-C chemokine receptor 5; CXCR3: CXCR5: C-X-C chemokine receptor 3; Tfh: T follicular helper; SLE: Systemic Lupus Erythematosus; ss: single-stranded.

### 3.2. B Cell Responses in the Setting of Mild COVID-19 Disease

Based on the current literature, SARS-CoV-2 specific memory B cells (measured by fluorescent tetramers) and bone-marrow resident plasma cells (measured by ELISpot) appear to be preserved for at least 7 months after infection. Quantitative measurements of memory B cells and plasma cells do appear to correlate significantly with virus-specific serum Ab titers and circulating Tfh cell numbers in individuals with mild COVID-19 disease [75,77].

### 3.3. B Cell Responses in the Setting of Severe COVID-19 Disease

Severe COVID-19 is characterized by a number of aberrations relating to the humoral immune compartment [102,103]. T-bet^+^ plasmablasts are notably expanded (these can account for >30% of the circulating B cells), as are activated naïve B cells (IgD^+^CD27^−^CD21^lo^CD11c^hi^) and atypical late transitional B cells (IgD^+^CD27^−^CD10^−^CD73^−^CXCR5^−^) [9,102]. Conversely, canonical naïve (IgD^+^CD27^−^), early transitional T1 and T2 (IgD^+^CD27^−^CD10^+^CD45RB^−^), CXCR5^+^ follicular (IgD^+^CD27^−^CD10^−^CD73^+^), isotype-unswitched (IgD^+^CD27^+^) and isotype-switched (IgD^−^CD27^+^) memory B cell subsets have been reported to be markedly diminished in the circulation [102]. In addition, sampling of secondary lymphoid tissues from patients that had succumbed to COVID-19 has revealed splenic white pulp atrophy, similar to observations in decedents that were infected with Ebola, Marburg and H5N1 influenza [102,107,108,109,110,111,112]. On a related note, patients infected with the 2002–2003 SARS-CoV also displayed disruption in the germinal center formation in sampled lymph nodes, and likewise, loss of Bcl6^+^ germinal center (GC) B cells is prominently featured in severe COVID-19 disease, although activation-induced cytidine deaminase (AID)^+^ B cells persist [102,113]. One subset of B cells that is conspicuously over-represented in patients that experience adverse outcomes following SARS-CoV-2 infection is the IgD^−^CD27^−^ (“double-negative”; DN) subset (Figure 2) [102,103]. These “disease-related” DN-B cells, that are also observed in chronic infections and autoimmunity, are not GC-derived, but do display T cell dependent isotype class-switching and fail to support the differentiation of Bcl6^−^Tfh cells into Bcl6^+^ Tfh cells, a step that is a pre-requisite for GC formation [102,103]. At least one report has also identified a DN B cell subset which is additionally lacking CD21 and CD11c that is unique to severe COVID-19 [103]. Although these disease-related, extra-follicular DN-B cells lack adequate CXCR5 expression, they often upregulate CXCR3 and therefore harbor a propensity to accumulate in tissues undergoing IFNγ (Th1)-mediated inflammation (Figure 2) [103]. Congruent with this observation are the findings that circulating Th1 effector T cells can be readily detected for at least 3 months even in individuals that have only experienced mild COVID-19, and that exaggerated TNFα levels can inhibit GC formation [74,114]. Severe COVID-19 is also associated with expansions of auto-antibody secreting B cells, such as the low mutation *IGHV4-34* Ab-secreting cells, which are also observed in the setting of systemic lupus erythematosus (SLE), suggesting disruptions in tolerogenic mechanisms coupled with a hyper-inflammatory milieu in a subset of COVID-19 patients [103]. Given that there is no conclusive evidence as yet that SARS-CoV-2 establishes chronic infections in immunocompetent hosts, the marked expansions of these DN cells could likely be the consequence of a dysregulated inflammatory axis in a subset of SARS-CoV-2 infected patients. In depth analyses of SARS-CoV-2 vaccine-induced B cell responses in patients that initially experienced infection-induced expansions of DN B cells could provide insights into the inflammatory cues that induce this specific DN-B response. Specifically, the inability to detect DN B cells following immunization with viral spike Ag vaccine constructs might implicate non-spike viral antigenic determinants in inducing this pathologic B cell subset. Conversely, detecting this DN B cell subset in a majority of vaccinated individuals that do not experience any vaccine-induced adverse clinical events, might call into question the direct pathologic role this subset is purported to possess in the setting of SARS-CoV-2 infection.

### 3.4. B Cell Responses in MIS-C

Interestingly, MIS-C is also associated with an elevation in plasmablast frequencies, despite the fact that this clinical entity is diagnosed 4–6 weeks following SARS-CoV-2 infection, suggesting either a delayed onset of or a prolonged plasmablast response [67]. Furthermore, plasmablasts in MIS-C patients express higher levels of the transcription factors T-bet and Eomesodermin when compared to pediatric COVID-19 patients with and without acute respiratory distress syndrome (ARDS) [67]. It is also worth noting that circulating plasmablast frequencies do not correlate with spike-receptor binding domain (RBD)-specific IgG and IgM in MIS-C, non-MIS-C or adult COVID-19 patients [67]. A similar lack of correlation between the circulating plasmablast response and total or activated Tfh cells is observed in MIS-C and non-MIS-C subjects [67].

### 3.5. B Cell Response in COVID-19 Vaccine Recipients

Individuals that have recovered from mild COVID-19 display a significant boost in NAb titers and memory B cell responses after the first dose of the vaccine, and these responses are not further amplified following the second dose of the vaccine administered three weeks after the first dose [101]. In contrast, SARS-CoV-2 naive individuals only amplify the NAb titers and memory B cell responses optimally after the second dose of the vaccine, suggesting adequate priming of the virus-specific immune response following mild COVID-19 [101]. The proximity in the timing of the first two vaccine doses likely contributes to this “plateauing” of the response in the infected→vaccinated group, since the memory B cells activated after the first dose of the vaccine may not have had enough time to reset and recharge to be optimally engaged in the tertiary immune response following the second dose of the vaccine in this group. Additionally, the prevailing evidence suggests no significant correlation between post-boost serum Ab levels and memory B cell responses in SARS-CoV-2 naive individuals, similar to what has been noted for some of the other licensed vaccines [80,101]. However, baseline memory B cell frequencies strongly correlate with boosted serum Ab levels but not baseline serum Ab levels in individuals that have recovered from mild COVID-19. Furthermore, induction of B cell memory does appear to be negatively impacted by advancing age [101]. Now that the third dose of the mRNA vaccine has received emergency use authorization (EUA) from the Food and Drug Administration (FDA) in the US for individuals ≥16 y of age that received the second dose of the vaccine at least 6 months prior, it will be worthwhile to measure and compare spike protein specific Ab levels after the third vaccine dose in the infected–vaccinated and uninfected–vaccinated groups [115]. The assumption would be that the magnitude of the spike protein-specific Ab responses in the two groups will be more or less equivalent; however, if a significant disparity is observed in this measurement between the two groups it would suggest that there may exist qualitative and quantitative differences in the spike-specific B cell memory pool that is induced by natural infection versus vaccination.

## 4. Evaluation of the SARS-CoV-2-Related Immune Response in the Immune-Suppressed State

Not surprisingly, the pandemic has amplified the risks and health-related concerns faced by individuals with compromised immune systems. Although Ab and T cell responses to SARS-CoV-2 are detected in patients that are immunosuppressed, the durability of these responses is clearly not on par with that observed in healthy individuals [116,117,118,119,120,121,122,123,124,125,126,127,128]. Reports have emerged that demonstrate that, in certain cases, even four doses of the SARS-CoV-2 vaccines may not elicit durable protective immunity [129,130]. Hence, in the absence of anti-viral drugs with proven long term safety and efficacy against SARS-CoV-2, repeated and frequent boosting with the vaccine, or the administration of SARS-CoV-2 specific NAbs will be the only viable options to effectively manage instances of viral persistence, as well as exposure to emerging variants of concern in immunosuppressed individuals [131,132].

## 5. Conclusions

No vaccine is 100% effective, and the intricate interplay between the host background genes with the molecular determinants of the host immune response to the pathogen dictates the complex nature of host–pathogen dynamics, and, collectively, this can and does influence the magnitude, longevity and the protective and/or pathologic disposition of the host immune response [51]. Hence, a thorough understanding of all relevant components of the immune response is warranted in the setting of infectious diseases, particularly where a novel pathogen displays evidence of pandemic spread and the potential for establishing endemicity. An ideal vaccine should induce a well-orchestrated immune response that appropriately calibrates and engages multiple components of the immune system. Given that the B cell derived NAbs target the extracellular virus, and the distinct likelihood that this response will minimize but not completely prevent infection, it would become incumbent on the T cell arm to deploy the precise clones targeting multiple viral antigenic determinants to purge the pathogen and induce sterilizing immunity. Such a sequential, coordinated and measured T cell response might be especially useful for optimizing protection for the host from both the pathogen directly (via providing help to B cells to generate NAbs), as well as from the immunopathology that might potentially be mediated by the cross-reactive and/or pre-existing memory T cell response driven by the residual antigen load. For instance, a sub-optimal, low titer NAb response might result in inadequate viral clearance following secondary infection, which could engender an over-exuberant secondary T cell response that might potentially contribute to immunopathology in some individuals that are predisposed to hyper-inflammatory responses. We have gained much in the way of knowledge about the immune response to SARS-CoV-2 in the past 24 months, but this will likely be just the tip of the iceberg moving forward as we learn to better contain or perhaps co-exist with this 21st century scourge.

## Figures and Tables

**Figure 1 cells-11-00067-f001:**
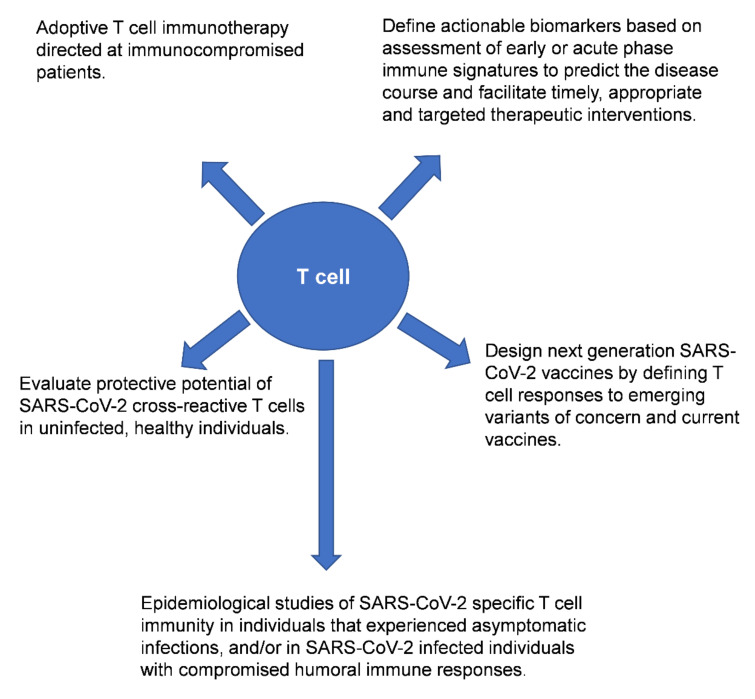
Potential downstream applications of profiling bulk or virus-specific T cells in the context of SARS-CoV-2 infection and/or after COVID-19 vaccination. Reference(s) [8,9,10,11,12,13,14,15,16,17,18,19,20,21,22,23,24,25,26,27,28,29,30,31,32,33,34,35,36,37,38,39,40,41,42,43,44,45,46].

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
