# Peer review of "Elucidating T Cell and B Cell Responses to SARS-CoV-2 in Humans: Gaining Insights into Protective Immunity and Immunopathology"

_cells, 2021, doi:10.3390/cells11010067_

Round 1
Reviewer 1 Report
The article entitled "Elucidating T cell and B cell responses to SARS-CoV-2 ... " by A. Khanolkar is a conventional clearly structured review that present some of the most interesting results about studies on the role of immune response in SARS-Cov2 infection. This review establish the two conventional immune pathways (T and B cells), but after this simple but correct general structure, my main concern as reviewer arrive when in the text an interlaced reading line is not followed, the article seems more like a "list" of results of main articles than the expected elaborate proposal, where no interpretation or own critical opinion of the data is expressed. Although there are two tables (the first one should be improved, because it is quite unclear why the authors just present 3 potential downstream applications of T-cell determination without explaining the reason of choosing these 3 between several other (p.e. better epidemiological studies, ...), no figures are presented in this article. For example, to better explain the SARS-CoV-2 specific T-cell therapy options (which are only very briefly cited, increasing the citation of articles on adoptive cell immunotherapy would be desirable), a figure would make it easier to visualize these options. Another figure about about the interrelationship between both pathways (T and B) will increase the attractiveness of the article.
As minor concerns:
- In relation to cross-reactive T cells, authors do not mention the article by L. Loyal et al., Science 10.1126 / science.abh1823 (2021), where the contribution of cross-reactive T cells both in natural infection and in post-vaccination response is clearly presented.
- Self-citations of 38 and 39 articles are unclearly related with the review and probably are inappropiated.
- In the section "Correlating T cell responses with COVID-19 disease phenotypes", factors associated with severe forms of the disease are not clearly mentioned. In this sense, it does not explain the dysregulation with the characteristic hyperinflammatory phase present in severe / critical cases, enhanced probably by adaptive immune response.
- Authors should do some mention of the option of using Treg therapy in severe (hyperinflammated) patients.
- Lack of comments on the immune responses in immunosuppressed patients (including natural infection and vaccines) is a main limitation of the paper.
- In "Conclusion" section, the sentence “As well as from the immunopathology that might potentially be mediated by the cross-reactive and/or pre-existing memory T cell response driven by the residual antigen load” is not so clear as it is stated here. The role of cross-reactive T cells is how much less discussed; some articles comment that they could have a protective role.
- More specifically:
- Line 92. Does not mention anything about anti-IFN antibodies as a serious COVID factor ( P. Bastard et al., Science 2020).
- Line 144 talks about Crotty et al and puts the citation 15 which is from another Moderbacher's article.
Reviewer 2 Report
Dear author,
The manuscript entitled (Elucidating T cell and B cell responses to SARS-CoV-2 in humans: gaining insights into protective immunity and immunopathology). is a well- written and interesting review which is beneficial to scientists during this pandemic.
Kindly add references to table 1 and 2. Some graphs or illustrations would make it more interesting.
Thanks
Round 2
Reviewer 1 Report
It is important to thank to the author the effort for trying to improve the manuscript following several comments of the reviewer's. Although all the concerns are not completely answered, the article is now more clear for the reading, specially after the introduction of the figures.